# Plastic Bending at Large Strain: A Review

Sergei Alexandrov [1,2] 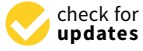, Elena Lyamina [1] and Yeong-Maw Hwang [3,*]

1    Laboratory for Technological Processes, Ishlinsky Institute for Problems in Mechanics RAS, 119526 Moscow, Russia; sergei_alexandrov@spartak.ru (S.A.); lyamina@inbox.ru (E.L.)

2    Faculty of Materials Science and Metallurgy Engineering, Federal State Autonomous Educational Institution of Higher Education "South Ural State University" (National Research University), 76, Lenin Prospekt, 454080 Chelyabinsk, Russia

3    Department of Mechanical and Electro-Mechanical Engineering, National Sun Yat-Sen University, Kaohsiung 80424, Taiwan

*    Correspondence: ymhwang@mail.nsysu.edu.tw

**Abstract:** Finite plastic bending attracts researchers' attention due to its importance for identifying material properties and frequent occurrence in sheet metal forming processes. The present review contains theoretical and experimental parts. The theoretical part is restricted to analytic and semi-analytic solutions for pure bending and bending under tension. The experimental part mainly focuses on four-point bending, though other bending tests and processes are also outlined.

**Keywords:** plastic bending; pure bending; bending under tension; semi-analytic solutions; bending tests



## 1. Introduction

All sheet metal forming processes incorporate some bending. A brief overview of the appearance of bending in sheet metal forming processes is provided in [1]. An approach based on determining the prevailing deformation path in combined bending and stretching to analyze real deep drawing processes was proposed and used in [2,3]. A similar approach was also adopted in [4]. The approaches apply to other real sheet metal forming processes and require a theoretical solution for the bending under tension process. The first part of the present paper reviews analytic and semi-analytic solutions for the pure bending and bending under tension processes of wide sheets at large strain. Almost all of these solutions are under plane strain conditions. Both elastic- and rigid-plastic material models are considered. No finite element and other numerical solutions are included in this review. One can use the theoretical solutions included as benchmark problems for verifying numerical codes, which is a necessary step before using such codes [5,6]. A review of analytic elastic/plastic solutions at infinitesimal strains is provided in [7]. Monograph [8] reviewed many theoretical solutions for non-plane strain bending, such as bending of beams, axisymmetric bending of circular plates, pressing rectangular plates into doubly-curved dies, and wrinkling of circular plates and flanges.

The second part of the present paper is devoted to an efficient method of finding analytic and semi-analytic solutions for the plane strain pure bending and bending under tension processes of sheets made of incompressible materials. Most of the contents of this part are based on the authors' research in this area.

The third part of the present paper reviews experimental works in which bending is used to identify the material properties.

## 2. Analytic and Semi-Analytic Solutions for Finite Bending of Wide Sheets

A schematic diagram of the bending process is shown in Figure 1, where *M* is the bending moment and *F* is the tensile force. Both are per unit width. Equilibrium demands that some pressure *P* is applied over the surface *CD*. This pressure is the reaction of a

tool. In the case of pure bending, $F = 0$ and $P = 0$. The solutions reviewed in this section assume that:

(i)     *AB* and *CD* are circular arcs throughout the process of deformation;
(ii)    *CB* and *AD* are straight lines throughout the process of deformation;
(iii)   End effects near the surfaces *CB* and *AD* are neglected;
(iv)    Pressure *P* is uniformly distributed over the surface *CD*;
(v)     Friction at the surface *CD* is neglected.

These assumptions suggest using a polar coordinate system $(r, \theta)$. The origin of this coordinate system, $O_p$, moves along the *x*-axis as the deformation proceeds. The neutral line is determined by the equation $r = r_n$ where $r_n$ is constant at each instant. The circumferential strain rate vanishes at $r = r_n$ (i.e., the material fiber coinciding with the neutral line undergoes no change of length in a further infinitesimally small strain).

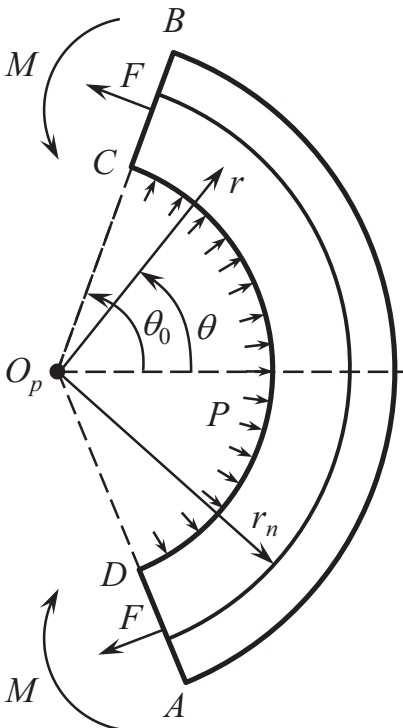

**Figure 1.** Schematic diagram of the bending process.

*2.1. Rigid/Plastic Solutions*

The first rigorous solution for both pure plane strain bending and plane strain bending under tension processes was proposed in [9] for isotropic rigid perfectly plastic materials. The constitutive equations comprise the yield criterion and its associated flow rule. The analysis starts with the solution for stress, and then, kinematic quantities are determined. The coordinate curves of the polar coordinate system (Figure 1) coincide with the trajectories of the principal stress directions. In this case, the yield criterion of any incompressible isotropic material that complies with the principle of maximum plastic work is expressed solely in terms of the stress variable $\sigma_r - \sigma_\theta$ where $\sigma_r$ is the radial stress and $\sigma_\theta$ is the circumferential stress. In particular,

$$\sigma_r - \sigma_\theta = 2k \quad \text{in the range} \quad r_{CD} \leq r \leq r_n, \tag{1}$$
$$\sigma_\theta - \sigma_r = 2k \quad \text{in the range} \quad r_n \leq r \leq r_{AB}.$$

Here, $k$ is the shear yield stress, $r_{AB}$ is the radius of the arc *AB*, and $r_{CD}$ is the radius of the arc *CD*. The shear yield stress is constant. Equilibrium demands that the radial stress is

continuous across the neutral line. Then, it is evident from (1) that the circumferential stress is discontinuous across this line. The solution [9] is entirely analytic. In the case of pure bending, its remarkable properties are that the radius of the neutral line is the geometric mean of $r_{AB}$ and $r_{CD}$, and the thickness of the sheet and bending moment do not change throughout the process of deformation. In particular,

$$\frac{M}{k} = \frac{H^2}{2}. \tag{2}$$

Here, $H$ is the initial thickness of the sheet.

This solution [9] was adopted in [10] for deriving a closed-form expression for strain at any fiber. This paper concluded that the basic assumptions made in [9], i.e., the assumptions shown at the beginning of this section, are plausible.

Paper [11] provided a semi-analytic plane strain solution for pure bending of anisotropic sheets. The corresponding isotropic solution is derived as a special case. The Runge–Kutta method is adopted for solving an ordinary differential equation. By assumption, the principal axes of anisotropy coincide with the coordinate curves of the polar coordinate system (Figure 1) throughout the process of deformation. This assumption becomes a consequence of the solution if the evolution of anisotropy follows the law proposed in [12], and the principal axes of anisotropy are parallel and perpendicular to the *x*-axis at the initial instant. Hill's quadratic yield criterion [9] was adopted in [11]. However, this assumption is superfluous for finding the general solution. The yield criterion of any incompressible anisotropic material that complies with the principle of maximum plastic work is expressed solely in terms of the stress variables $\sigma_r - \sigma_\theta$ and the shear stress in the polar coordinate system [13]. Since the shear stress vanishes, the yield criterion involves $\sigma_r - \sigma_\theta$ only. However, one needs to choose a specific yield criterion for getting quantitative results. Paper [11] also attempted to account for isotropic and kinematic hardening. These hardening laws involve the equivalent plastic strain. However, the equivalent plastic strain, which is calculated by integrating over the actual strain path, is replaced with the total strain found using the initial and final configurations. These quantities have the same numerical value only if there is no strain reversal. Meanwhile, a strain reversal occurs in a material fiber that momentarily coincides with the neutral line. The solution provided in [14] is free of this drawback. In this paper, a procedure for calculating the correct value of the equivalent plastic strain in a region where a strain reversal occurs is developed. An iterative procedure in conjunction with a numerical method for solving an ordinary differential equation is required for getting quantitative results. The analysis in [14] was limited to Swift's law in the case of isotropic hardening. In addition to homogeneous sheets, bonded laminated metals are considered. In total, five different sheets are studied in detail. Those are: (1) a non-strain hardening bi-metal sheet, (2) a non-strain hardening tri-metal sheet, (3) a strain hardening mono-metal sheet, (4) a strain hardening mono-metal sheet, accounting for a simplified Bauschinger effect, and (5) a strain hardening bi-metal sheet.

A simplified method to deal with rigid/plastic hardening materials is to adopt an artificial strain distribution in the region between the current neutral line and the original center fiber [15].

Analytic and semi-analytic solutions for finite bending of various hydrogel strips were found in [16–20]. An elastomer-hydrogel bi-layer strip was considered in [16], a hydrogel-elastomer-hydrogel tri-layer strip in [17], and a functionally graded hydrogel strip in [18]. The solution given in [19] is for temperature-induced bending of a bi-layer gel strip. An analytic method was developed and used in [20] for analyzing the swelling-induced bending process of a bi-layer strip made of a pH-sensitive layer attached to an inert elastomer layer.

### 2.2. Elastic/Plastic Solutions

Elastic/plastic analysis is much more complicated than rigid/plastic analysis. One of the main difficulties is that the plane strain assumption does not imply that both the elastic

and plastic portions of the corresponding strain vanish unless the material is incompressible. Another possible case when both portions vanish may occur if the material obeys Tresca's yield criterion. Therefore, available analytic and semi-analytic solutions often involve additional assumptions.

Several solutions are based on the assumption, in addition to the assumptions formulated at the beginning of Section 2, that the radial stress in the polar coordinate system shown in Figure 1 vanishes. An oversimplified solution for the bending under tension process based on the assumption of the linear through-thickness distribution of strain was provided in [21]. The material obeys Ludwik's hardening law. The solution presented in [22] ignores the existence of the stretching, compressing, and strain reversal areas in the process of bending under tension. The total strain is found using the initial and final configurations. Therefore, this approach is similar to that developed in [11] for rigid/plastic models. The non-quadratic Hill's yield criterion proposed in [23] was used in [22]. The equivalent stress follows Swift's hardening law. Paper [22] emphasized thickness variation and springback. The approach proposed in [22] was extended to repeating bending, unbending, and reverse bending in [24]. Cyclic material models were adopted. One can find numerous solutions based on the assumption that the radial stress vanishes in the literature, for example [25–30]. Paper [31] introduced a new methodology for springback prediction and then used it to analyze the cyclic bending process. An advantage of this methodology is that it is purely analytic for springback prediction. A comparison with the methodology proposed in [32] was made.

The deformation theory of plasticity was employed in [33] to describe the pure bending process. The constitutive equations were based on the Hencky strain and the Cauchy stress. The elastic linear hardening material model was adopted. Analytic solutions were found in two plastic regions and an elastic region. The plastic regions propagate from the outside and inside surfaces of the sheet, and the elastic region is located between the plastic regions. A combination of these solutions provides the solution for the entire sheet. Several particular solutions for the entire sheet were derived in [33]. Paper [34] provided a solution for an incompressible isotropic nonlinearly elastic material to investigate the effect of strain stiffening. The stresses and bending moments were obtained explicitly for several strain-energy densities suitable for rubber-like materials. Another solution for the deformation theory of plasticity was found in [35], where the material was supposed to be elastic-plastic, power-law hardening. The flow theory of plasticity in conjunction with a numerical method was also employed in this paper.

A unique solution for the pure bending process is found in [36], where elastic compressibility and a constant change of the sheet's width were taken into account. The Eulerian rate-type material elastoplastic model accounts for both isotropic and kinematic hardening. It is based on Tresca's yield criterion and its associated flow rule.

## 3. Efficient Method for the Analysis of Bending

Most of the contents of this section are based on the authors' research devoted to developing and using an efficient method for the analysis of plane strain bending of incompressible sheets. The method was proposed in [37] for pure bending and extended to bending under tension in [38]. Figure 2 illustrates the initial and final configurations together with several coordinate systems.

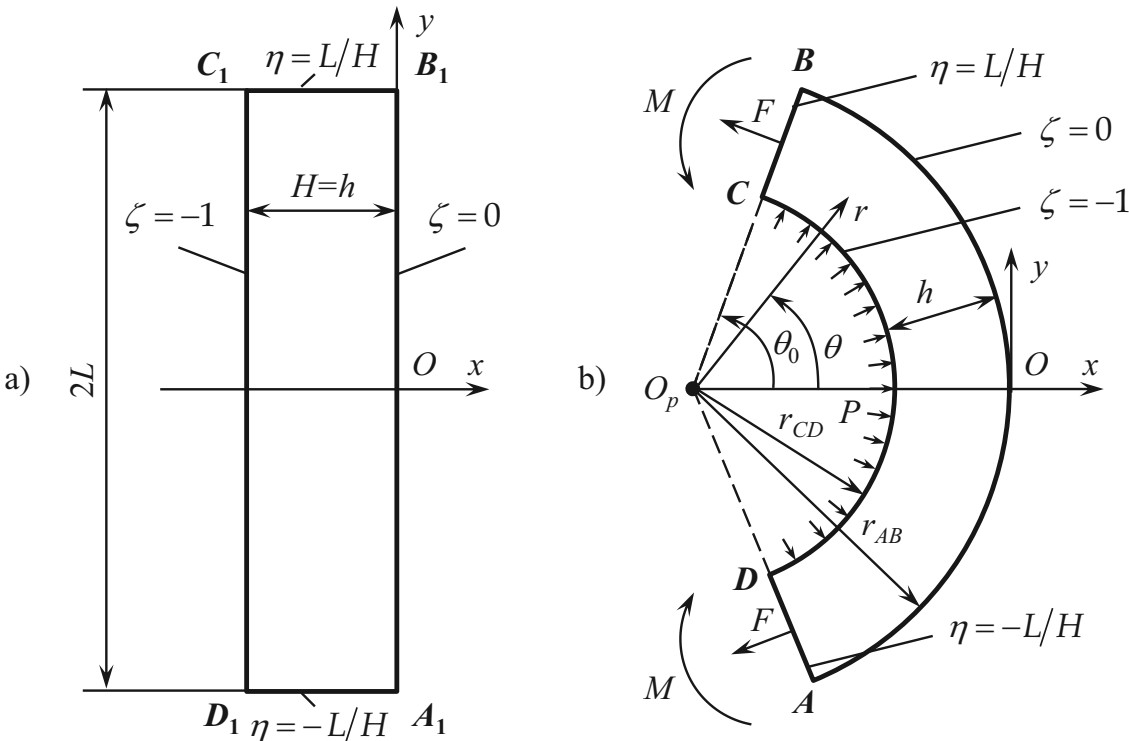

**Figure 2.** Schematic diagram of the bending process: (**a**) initial configuration; (**b**) intermediate and final configurations.

A starting point of the method is the mapping between Eulerian Cartesian coordinates $(x, y)$ and Lagrangian coordinates $(\zeta, \eta)$ in the form:

$$\frac{x}{H} = \sqrt{\frac{\zeta}{a} + \frac{s}{a^2}} \cos(2a\eta) - \frac{\sqrt{s}}{a} \quad \text{and} \quad \frac{y}{H} = \sqrt{\frac{\zeta}{a} + \frac{s}{a^2}} \sin(2a\eta). \tag{3}$$

Here, $a$ is a time-like variable such that $a = 0$ at the initial instant and $s$ is a function of $a$. The latter should be found from the solution. The $x$-axis is an axis of symmetry of the process. The Cartesian coordinate system's origin is situated at the intersection of the axis of symmetry and curve $AB$. At the initial instant,

$$x = \zeta H \quad \text{and} \quad y = \eta H. \tag{4}$$

Using l'Hospital's rule, one can verify that this condition is satisfied if:

$$s = \frac{1}{4}. \tag{5}$$

at $a = 0$. As mentioned at the beginning of Section 2, it is convenient to use the polar coordinate system $(r, \theta)$. The origin of this coordinate system is located at $x = -H\sqrt{s}/a$ and $y = 0$. It follows from (3) that:

$$\frac{r}{H} = \frac{\sqrt{\zeta a + s}}{a} \quad \text{and} \quad \theta = 2a\eta. \tag{6}$$

It is seen from (4) and Figure 2 that $\zeta = 0$ on curve $AB$ and $\zeta = -1$ on curve $CD$ throughout the process of deformation. The radii of circular arcs $AB$ and $CD$ are determined from (6) as:

$$\frac{r_{AB}}{H} = \frac{\sqrt{s}}{a} \quad \text{and} \quad \frac{r_{CD}}{H} = \frac{\sqrt{s-a}}{a}, \tag{7}$$

respectively. Using a standard procedure, one can verify that the Lagrangian coordinate system is orthogonal, and the shear strain rate referring to this system vanishes throughout the

process of deformation. The latter means that the shear stress referring to the Lagrangian coordinate system vanishes throughout the deformation process if the material model is coaxial (i.e., the principal directions of the stress and strain rate tensors coincide). In what follows, it is assumed that this condition is satisfied. The mapping (3) is independent of the constitutive equations. Using this equation, one can calculate the components of the total strain and strain rate tensors. In particular, it is possible to verify that the equation of incompressibility is satisfied. The following equation determines the neutral line:

$$\zeta = \zeta_n = -\frac{ds}{da}. \tag{8}$$

The total principal strains are:

$$\varepsilon_\zeta = -\frac{1}{2}\ln[4(\zeta a + s)] \quad \text{and} \quad \varepsilon_\eta = \frac{1}{2}\ln[4(\zeta a + s)]. \tag{9}$$

Those are also the normal strains in the Lagrangian coordinates. The strain rate components referring to these coordinates are:

$$\xi_\zeta = -\frac{(\zeta + ds/da)}{2(\zeta a + s)}\frac{da}{dt} \quad \text{and} \quad \xi_\eta = \frac{(\zeta + ds/da\ )}{2(\zeta a + s)}\frac{da}{dt}. \tag{10}$$

Here, $t$ is time. The equivalent strain rate involved in viscoplastic models is readily determined from (10).

The stress solution exists if the mapping (3) is compatible with the equilibrium and constitutive equations. Let $\sigma_r$ and $\sigma_\theta$ be the radial and circumferential stresses in the polar coordinate system, respectively. The only stress equilibrium equation that is not identically satisfied in the polar coordinate system is:

$$\frac{\partial \sigma_r}{\partial r} + \frac{\sigma_r - \sigma_\theta}{r} = 0. \tag{11}$$

It is evident from (6) that $\sigma_r = \sigma_\zeta$ and $\sigma_\theta = \sigma_\eta$, where $\sigma_\zeta$ and $\sigma_\eta$ are the normal stresses in the Lagrangian coordinate system. One can replace differentiation with respect to $r$ with differentiation with respect to $\zeta$ using (6). Then, Equation (11) becomes:

$$\frac{\partial \sigma_\zeta}{\partial \zeta} + \frac{a(\sigma_\zeta - \sigma_\eta)}{2(\zeta a + s)} = 0. \tag{12}$$

One of the constitutive equations is the yield criterion. A wide class of materials obeys the yield criterion of the form:

$$|\sigma_\eta - \sigma_\zeta| = \sigma_0 \Lambda\left(\varepsilon_{eq}^p, \xi_{eq}^p\right). \tag{13}$$

Here, $\sigma_0$ is a reference stress, $\varepsilon_{eq}^p$ is the equivalent plastic strain, $\xi_{eq}^p$ is the equivalent plastic strain rate, and $\Lambda$ is a known function of its arguments. Various definitions of the equivalent strain and the equivalent strain rate are possible. Independent of the specific definitions, one can express the right-hand side of Equation (13) as a function of $a$, $s$, $\zeta$, and $ds/da$ using the kinematic relations above and Hooke's law. The latter is not required in the case of rigid/plastic models. Equations (12) and (13) combine to give:

$$\frac{\partial \sigma_\zeta}{\partial \zeta} \pm \frac{a\Lambda\left(\varepsilon_{eq}^p, \xi_{eq}^p\right)}{2(\zeta a + s)} = 0. \tag{14}$$

The upper sign corresponds to the region $-1 \leq \zeta \leq \zeta_n$ and the lower sign to the region $\zeta_n \leq \zeta \leq 0$. In general, Equation (14) should be solved numerically, and we leave the general analysis at this point, with the solution clear in principle.

Further simplifications are possible for particular functions $\Lambda\left(\varepsilon_{eq}^p, \xi_{eq}^p\right)$. In the case of a rigid perfectly plastic material, $\Lambda\left(\varepsilon_{eq}^p, \xi_{eq}^p\right)$ is constant. Therefore, Equation (14) can

be immediately integrated. The solution eventually reduces to the solution given in [9]. The function $\Lambda\left(\varepsilon_{eq}^{p}, \xi_{eq}^{p}\right)$ is independent of $\xi_{eq}^{p}$ for rigid plastic hardening materials. The corresponding solution for pure bending is found in [37]. The function $\Lambda\left(\varepsilon_{eq}^{p}, \xi_{eq}^{p}\right)$ is independent of $\varepsilon_{eq}^{p}$ for rigid viscoplastic materials. The corresponding solution for pure bending is also found in [37].

It was shown in [38] that the solution for elastic/plastic strain hardening materials is facilitated using the equivalent plastic strain as an independent space variable instead of $\zeta$ in plastic regions. In particular, Equation (14) becomes:

$$\frac{\partial \sigma_{\zeta}}{\sigma_0 \partial \varepsilon_{eq}^{p}} = \left(1 + \frac{\sigma_0 \Lambda'}{3G}\right)\Lambda. \tag{15}$$

Here, $G$ is the shear modulus of elasticity and $\Lambda' \equiv d\Lambda/d\varepsilon_{eq}^{p}$. It is worth reminding that $\Lambda$ is a function only of $\varepsilon_{eq}^{p}$ for this material model. Therefore, Equation (15) can be immediately integrated. This change of variables is useful for solving a class of elastic/plastic boundary value problems [39–41]. Paper [38] presented the solution for bending under tension. The corresponding solution for pure bending was provided in [42].

In addition to the solutions in [37,38,42], the method above has been employed for finding solutions for several other material models. A model of anisotropic plasticity was adopted in [43]. Paper [44] dealt with a model that accounts for Bauschinger's effect. The bending process of bi-layer sheets was studied in [45]. Each layer is elastic perfectly plastic.

Papers [46,47] extended the method above to sheets with continuously varying through-thickness properties. The material model adopted in [46] is isotropic. Paper [47] generalized this solution to anisotropic materials. The solution in [46] is for elastic/plastic materials. Both elastic/plastic and rigid/plastic models were considered in [47].

The evolution of damage in the pure bending process was investigated in [48] using a rigid/plastic model. The damage evolution equation adopted in this paper has the form:

$$\frac{dD}{dt} = \Omega\left(\frac{\sigma}{\sigma_{eq}}, \varepsilon_{eq}, D\right)\xi_{eq}. \tag{16}$$

Here, $d/dt$ denotes the convected derivative, $\sigma$ is the hydrostatic stress, $\sigma_{eq}$ is the equivalent stress, and $D$ is the damage parameter. Damage evolution equations of this type in conjunction with rigid plasticity are widely used in damage mechanics [49–52]. An advantage of the method above is that the convected derivative involved in (16) is identical to the local time derivative in the $(\zeta, \eta)$-coordinate system. However, the direct use of the method is impossible because it is necessary to add the damage parameter to the arguments of the function $\Lambda$ involved in (13) and (14). One should solve Equations (14) and (16) simultaneously. This system is hyperbolic, and a simple finite difference method solves it efficiently.

Ideal flows in plasticity were discovered in [53]. Non-steady ideal membrane flows have broadly been used as the basis of an inverse method for designing sheet forming processes, for example [54,55]. The theory of non-steady planar bulk ideal plastic flows was developed in [56]. Another advantage of using the $(\zeta, \eta)$-coordinate system is that one can efficiently design bending, considered as a bulk process, using the ideal flow theory. The condition for the bending under tension process to be an ideal flow process is that the neutral line is fixed in the material. Therefore, $\zeta_n$ should be independent of $a$. Then, it follows from (5) and (8) that:

$$s = -a\zeta_n + \frac{1}{4}. \tag{17}$$

This ideal flow solution was discussed in detail in [57]. It is the only non-trivial non-steady bulk ideal flow solution to the best of the authors' knowledge.

## 4. Comparison of Elastic/Plastic and Rigid/Plastic Solutions

In general, rigid plasticity is suitable for obtaining solutions at large strains in domains where the elastic strain rate components are much smaller than the plastic strain rate components. However, there are two issues in the case of bending. Firstly, an elastic region inevitably occurs in the pure bending process. Such a region also occurs in the bending under tension process if the tensile force is small enough. The plastic strain rate components vanish in this region, and the condition above is not satisfied. On the other hand, it is reasonable to expect that the thickness of the elastic region is very small as compared to the thickness of the sheet after a certain amount of deformation. Therefore, the effect of this region on the distribution of stress may be negligible. Secondly, the prediction of springback is critical in analyzing the bending process, which assumes using elastic/plastic solutions. This issue can be resolved using the approach proposed in [58]. According to this approach, the distribution of stresses found from a rigid/plastic solution at the end of the loading stage is used as the initial distribution for the stage of unloading, which allows one to calculate the distribution of residual stresses and strains. Paper [47] explored the accuracy of the approach proposed in [58] for the analysis of bending.

The method described in the previous section was used in [47] for obtaining elastic/plastic and rigid/plastic solutions for the pure bending of anisotropic sheets with a non-uniform through-thickness distribution of material properties. The quantitative results below are for the distribution of properties given in [59]. Figure 3 shows the variation of the locations of the elastic/plastic boundaries and the neutral line with $H/r_{CD}$ at the beginning of the process.

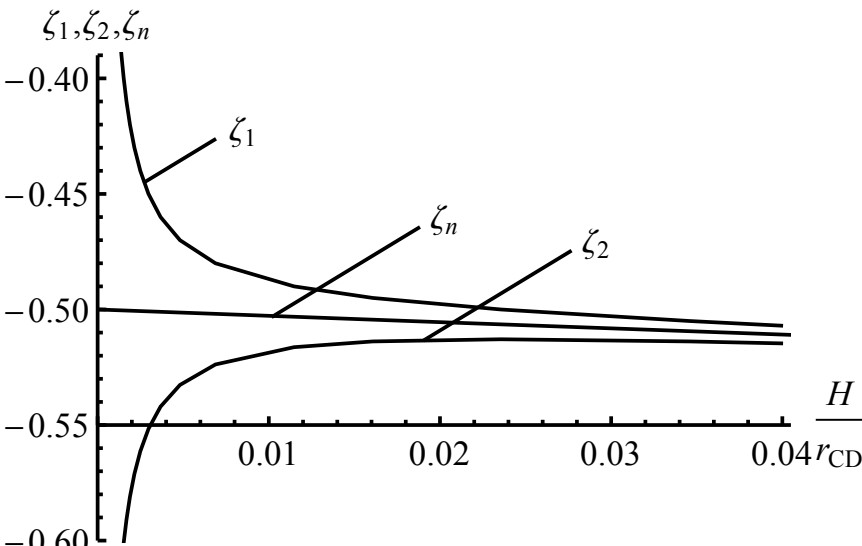

**Figure 3.** Variation of the locations of the elastic/plastic boundaries and the neutral line with $H/r_{CD}$ at the beginning of the process.

In this figure, $\zeta_1$ and $\zeta_2$ are the $\zeta$-coordinates of two elastic/plastic boundaries found from the elastic/plastic solution, and $\zeta_n$ is the $\zeta$-coordinate of the neutral line found from the rigid/plastic solution. Therefore, the elastic region in the elastic/plastic solution is $\zeta_2 \leq \zeta \leq \zeta_1$. It is seen from Figure 3 that the thickness of this region becomes small at the very beginning of the process. The through-thickness distribution of the radial and circumferential stresses is depicted in Figures 4 and 5, respectively.

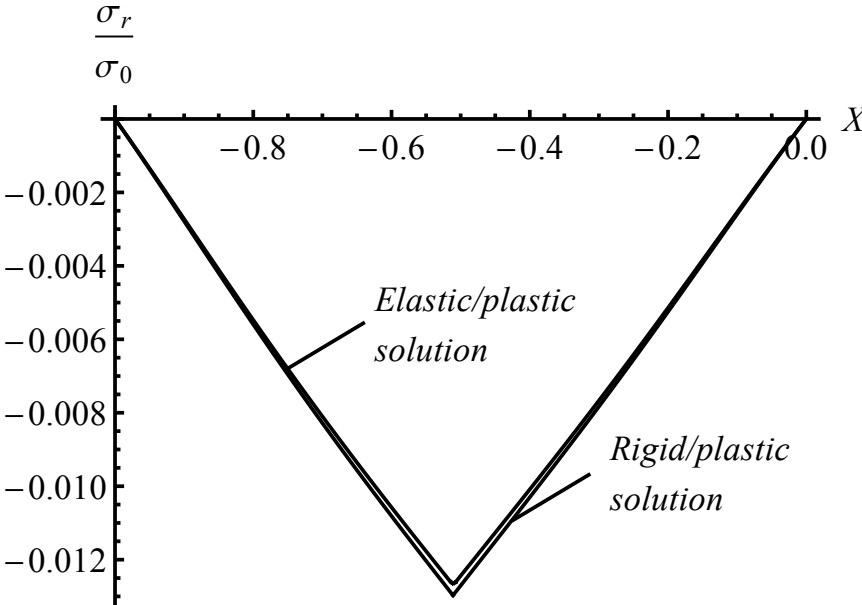

**Figure 4.** Distribution of the stress $\sigma_r$ at $H/r_{CD} = 0.04$ found using the elastic/plastic and rigid/plastic solutions.

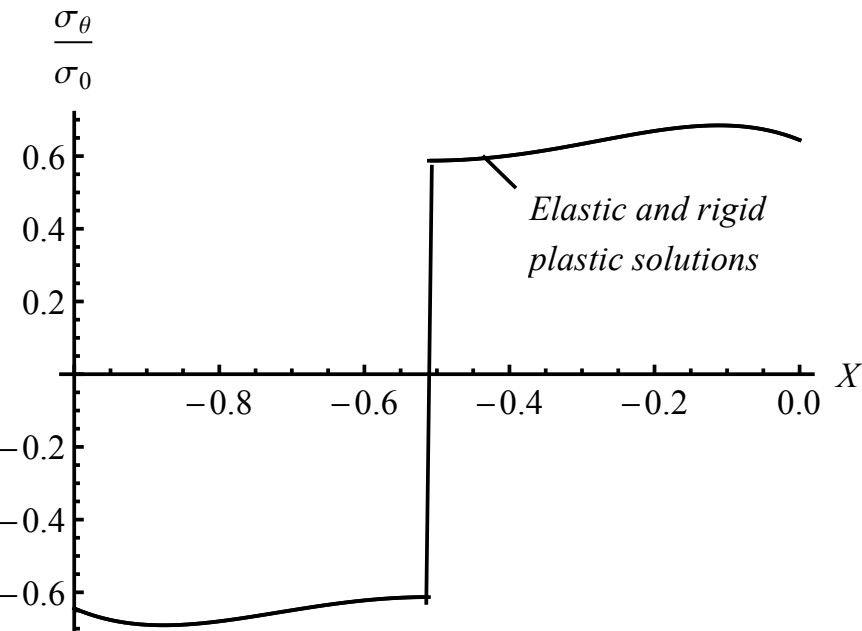

**Figure 5.** Distribution of the stress $\sigma_\theta$ at $H/r_{CD} = 0.04$ found using the elastic/plastic and rigid/plastic solutions.

In both figures,

$$X = \frac{r - r_{CD}}{H}. \tag{18}$$

The curves in Figures 4 and 5 correspond to $H/r_{CD} = 0.04$. It is seen from Figure 4 that the difference between the elastic/plastic and rigid/plastic solutions for the radial stress is very small and is visible only near the neutral line. The difference between the elastic/plastic and rigid/plastic solutions for the circumferential stress is invisible in Figure 5. Since the circumferential stress completely controls the bending moment, it is evident that the effect of the replacement of the elastic/plastic model with the rigid plastic one on its predicted magnitude is negligible. It is also clear from Figures 4 and 5 that the approach proposed in [58] is undoubtedly accurate for determining the distribution of

residual stresses in the process of bending, followed by unloading, if $H/r_{CD} \geq 0.04$ (i.e., the inside radius is 25 times larger than the initial thickness of the sheet).

## 5. Bending Test

A great deal of information is available in the literature regarding various bending tests. The review in this section is by no means comprehensive. It focuses on the mechanics of processes rather than on the materials science aspects.

In the pure bending process, the absolute value of the radial stress in the polar coordinate system (Figure 1) is much smaller than the absolute value of the circumferential stress. The state of stress is regarded as compressive in the region between the neutral line and the inside surface and as tensile in the region between the neutral line and the outside surface. Given these two regions' existence in one process, it is natural that the four-point bending test has attracted the attention of researchers for determining the stress-strain curves in tension and compression using a single test. According to [60], Herbert developed a corresponding method in 1910. Herbert published his paper in German, and it is not easy to access. However, paper [60] reproduced Herbert's method in detail. Moreover, this paper presented experimental data for three materials: beryllium, cast iron, and copper. A four-point bend apparatus was used. The tensile and compression stress-strain curves do not coincide. Comparing the stress-strain curves obtained from the bending test, conventional uniaxial tension test, and conventional uniaxial compression shows no significant difference. Of course, tension-compression asymmetry and other material properties can be studied using a three-point bending test, for example [61,62]. However, in this case, using numerical simulation for finding accurate solutions is unavoidable. The stress-strain curve for annealed copper was found in [63] using the method described in [60]. The limitations of the method were discussed in detail. In [64], the method was successfully used for obtaining the stress-strain curve of cement-based composites. Paper [65] modified the theoretical analysis described in [60] by replacing the original equations in incremental form with several ordinary differential equations, which simplifies using the method in practice. The stress-strain curves were determined for pure magnesium and S45C steel. The paper concluded that the calculation is sensitive to small changes of the bending curves' slope, which may lead to difficulty with solving the inverse problem. In [66], the method [65] was further developed and used to quantify Bauschinger's effect. The theoretical part of this modified method is based on finite element analysis. Paper [67] validated the strength differential effect of DP 980 steel from uniaxial tension-compression tests using the pure bending test carried out on a novel test apparatus.

The four-point bending test was adopted in [68] for investigating tension-compression flow stress asymmetry in NiTi shape memory alloy strips of less than 0.8 mm in thickness. The theoretical part of this research is based on the crystallographic theory of martensite. Though not the four-point bending test, the bending test is also useful for studying the material properties of foils. In [69], pure aluminum foil with thickness ranging from 25 μm to 500 μm were tested for investigating the springback behavior. The theoretical part of this research is based on a strain gradient theory of plasticity.

The four-point bending test can also be used to quantify the effect of the difference between the tension and compression stress-strain curves on the accuracy of finite element simulations [70]. Twin-roll cast, rolled, and annealed AZ31 sheets of different thickness were tested at different temperatures to determine the tension and compression stress-strain curves. The approach consists of comparing the solutions based on (a) the flow curve obtained from tensile tests, (b) the flow curve obtained from compression tests, (c) the flow curve obtained from tensile tests in the tensile region, and (d) the flow curve obtained from compression tests in the compressive region. The paper concluded that only the latter assumption allows predicting the maximum bending force with a deviation of less than 7% for all experiments.

The bending/unbending test allows investigating a larger spectrum of material properties than the bending test. Large uniaxial Bauschinger effects were quantified in [71] using

the pure bending/unbending test and interferometer techniques. Three materials were tested: DP 590 steel, DP 780 steel, and AA 6022 aluminum alloy. The bending-unbending test is also used for studying the evolution of microstructure, for example [72]. In this work, pure titanium sheets were tested. An inverse technique was proposed in [73] for identifying a nonlinear kinematic hardening law for anisotropic metals. Mild steel and two aluminum alloys were used in the experiment. Springback and its accurate theoretical prediction are crucial issues for processes that involve bending [74]. Paper [75] employed a multiple bending-unbending process for understanding the springback phenomenon. The process of continuous plastic bending under tension (CBT) is a multiple bending-unbending process. The CBT process was proposed in [76] for preventing neck formation in the tension testing of sheet materials. The test was carried out on various ferrous and nonferrous sheet specimens, and the total elongation increased from 30 to 1000%. The CBT process was further developed in [77–80]. A CBT machine for evaluating the formability of automotive aluminum alloy and advanced high strength steel sheets was designed and fabricated in [78]. The machine was used to test AA 6022-T4 in [78,80]. The evolution of microstructure in the same alloy was studied in [79]. Paper [77] proposed a method for identifying post-necking strain hardening behavior using the CBT process and finite element simulation. The method was successfully applied to AA 6022-T4 alloy, DP 980 steel, and DP 1180 steel.

From the theoretical perspective, the four-point bending test is an optimal bending test for determining the stress-strain curve. However, it is difficult to carry out this test when large displacements and rotations are required [81]. Therefore, many researchers have attempted to design devices that allow one to bend sheets under nearly pure bending conditions to sufficiently large strains. Several successful designs can be found in [82–85].

Contact friction and pre-strain affect the interpretation of experimental data. Friction unavoidably occurs in some bending tests. It is usually undesirable, and it is challenging to take it into account in the analysis of experimental data. Paper [86] developed a bending under tension test with direct friction measurement. The effect of lubrication on friction in the bending under tension test was studied in [87]. Pre-strain is a positive factor. One can induce a controllable amount of pre-strain before the bending test. This method allows obtaining additional data using the bending test. For example, Bauschinger's effect in hot rolled and aged steel was investigated in [88] and in interstitial free steel in [89] using the method above. The four-point bending test was used in [89]. The state of stress in the test conducted in [88] is also close to that in the pure bending process.

## 6. Conclusions

The present paper reviews research related to plastic bending. Its theoretical part concerns analytic and semi-analytic solutions for wide sheets. The material models considered are coaxial. All the solutions included in the review can be divided into two groups: (1) pure bending and (2) bending under tension. End effects are neglected. The pure bending process should closely reproduce the state of stress and strain in the four-point bending test. Therefore, this test is advantageous for identifying material properties. The experimental part of the present paper demonstrates several applications of the four-point bending test. Other bending tests and processes are also outlined in this part.

The considered material models fall into two categories: (1) rigid/plastic models and (2) elastic/plastic models. It is shown that the difference between the corresponding elastic/plastic and rigid/plastic solutions is negligible except at the very beginning of the process. Since the rigid/plastic solutions are much simpler, it is recommended to use them for bending at large strains. It is possible even if the distribution of residual stresses is required.

**Author Contributions:** Writing, S.A., Y.-M.H., and E.L. All authors have read and agreed to the published version of the manuscript.

**Funding:** This research was made possible by the grants RFBR-19-51-52003 (Russia), MOST 108-2923-E-110-002-MY3 (Taiwan), and AAAA-A20-120011690136-2 (Russia).

**Institutional Review Board Statement:** Not applicable.

**Informed Consent Statement:** Not applicable.

**Data Availability Statement:** Not applicable.

**Conflicts of Interest:** The authors declare no conflict of interest.

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
