# Peer review of "Plastic Bending at Large Strain: A Review"

_processes, doi:10.3390/pr9030406_

Round 1
Reviewer 1 Report
No suggestions or comments to the authors
Author Response
This reviewer recommends publication
Reviewer 2 Report
The manuscript entitled Plastic Bending at Large Strain: A Review written by S. Alexandrov et al. was presented for a review. It should be emphasized that the subject of the paper is interesting and relevant from both practical and scientific points of view. A review is a special type of the paper, where the Authors should show that they are specialists in this subject. This cryterion was fulfilled. The Authors present own (and not only) achievements in the field of bending. The problem of bending at large strain is described from theoretical and experimental points of view. The analytical solutions are clearly presented and explained. In the last part, where they show examples of bending tests, I feel some insufficiency. The Authors explain that they focus only on the mechanics of processes. However, some presentation of at least main achievements related to materials properties e.g. structure, hardness defects. etc. in the context of plastic bending at large strain would be welcome. In any case the paper is well written, is interesting and contains a rich literature (also published by the Authors). It will be a strong position in this kind of literature. I have no doubt, that it is proper for publication.
Author Response
This reviewers recommends publication
Reviewer 3 Report
Please follow my minor critical remarks listed in the fille attached to this message.

Author Response
We have incorporated all the comments into the revised version of the manuscript.